# A Case Study on the Behavioural Effect of Positive Reinforcement Training in a Novel Task Participation Test in Göttingen Mini Pigs

**DOI:** 10.3390/ani11061610

**Published:** 2021-05-29

**Authors:** Lisa Jønholt, Cathrine Juel Bundgaard, Martin Carlsen, Dorte Bratbo Sørensen

**Affiliations:** 1Department of Veterinary and Animal Sciences, Faculty of Health and Medical Sciences, University of Copenhagen, Gronnegaardsvej 15, 1870 Frederiksberg C, Denmark; li.joenholt@gmail.com; 2Novo Nordisk A/S, Novo Nordisk Park 1, 2760 Maalov, Denmark; cjb@corit.dk (C.J.B.); macn@novonordisk.com (M.C.)

**Keywords:** laboratory pigs, clicker training, positive reinforcement training, welfare

## Abstract

**Simple Summary:**

In laboratory animal research, many procedures and tests will be stressful for the animals, as they are forced to participate. Training animals to voluntarily participate using reward-based training such as clicker training or luring may reduce levels of stress, and thereby increase animal welfare. Clicker training is traditionally used in zoos, aquariums, and with pets to train the animals to cooperate during medical procedures, whereas in experimental research, luring seems to be the preferred training method. This descriptive case study aims to present the behaviour of clicker trained and lured pigs when they are subjected to a potentially fear- and stress-evoking behavioural test—the novel task participation test—in which the pigs must walk a short runway with a novel walking surface. All eight trained pigs voluntarily participated and only one of the lured pigs showed a behaviour indicating decreased welfare. Hence, training pigs to cooperate during experimental procedures resulted in a smooth completion of the task with no signs of fear or anxiety in seven out of eight animals, and we thus suggest that training laboratory pigs prior to experimental procedures or tests should always be done to ensure low stress levels.

**Abstract:**

In laboratory animal research, many procedures will be stressful for the animals, as they are forced to participate. Training animals to cooperate using clicker training (CT) or luring (LU) may reduce stress levels, and thereby increase animal welfare. In zoo animals, aquarium animals, and pets, CT is used to train animals to cooperate during medical procedures, whereas in experimental research, LU seem to be the preferred training method. This descriptive case study aims to present the behaviour of CT and LU pigs in a potentially fear-evoking behavioural test—the novel task participation test—in which the pigs walked a short runway on a novel walking surface. All eight pigs voluntarily participated, and only one LU pig showed body stretching combined with lack of tail wagging indicating reduced welfare. All CT pigs and one LU pig displayed tail wagging during the test, indicating a positive mental state. Hence, training pigs to cooperate during experimental procedures resulted in a smooth completion of the task with no signs of fear or anxiety in seven out of eight animals. We suggest that training laboratory pigs prior to experimental procedures or tests should be done to ensure low stress levels.

## 1. Introduction

Laboratory animals are often subjected to a large variety of environmental stressors, which compromises animal welfare and negatively affects animal physiology and psychology, thereby risking the quality of the research data [1,2,3,4]. Major sources of stress are experimental procedures, which may include painful or fear-evoking events. Often the animals are restrained or in some way forced to participate, and hence they cannot escape or avoid the stressors. Training the animals using positive reinforcement to participate voluntarily in experimental procedures such as drug dosing, injections, clinical examinations, or potentially fear-evoking behavioural tests reduces the stress experienced by the animal [5]. Using animal training to reduce stress and fear during medical and veterinary procedures was originally introduced in zoos and aquariums, where large and potentially dangerous animals were hazardous or difficult to draw blood samples from or anesthetize, for example [6]. In laboratory animal science, positive reinforcement training has been used in primates for decades, and results confirm that the animals are less stressed when they have been trained for the procedures [5,7,8].

Animal training using operant conditioning—and more specifically conditioned positive reinforcement—was popularized by B.F Skinner and later by Karen Pryor [9,10,11]. The basic principles of operant conditioning have been defined elsewhere [12,13,14,15,16]. In short, positive reinforcement training is a technique where a specific behaviour is reinforced, by adding a preferred stimulus like food, when the animal shows that behaviour. Hence, the consequence of showing that behaviour is pleasant and the likelihood of the animal showing that behaviour again is increased. In practical animal training, several techniques build on positive reinforcement (PR), such as luring, shaping, and targeting [17,18,19]. Technically, all of these positive reinforcement techniques can be combined with a conditioned (or secondary) reinforcer and is then called conditioned positive reinforcement or clicker training. Clicker training (CT) builds on a learned (conditioned) association between a previously neutral stimulus such as a whistle or a clicker (a small device that, when pressed, gives a distinct “click-click” sound) and a primary (or unconditioned) reinforcer, typically food. When the animal shows the correct response, the conditioned reinforcer is presented, followed by the primary reinforcer [16,20]. This training approach will improve the trainer’s timing as the behaviour desired by the trainer can be marked with high precision using the clicker. Clicker training has been applied over the years to a large number of animal species, ranging from fish and reptiles to elephants, dolphins, and orcas, training animals for shows, commercials, and husbandry procedures—including medical procedures [10,17,21]. Clicker training is also increasingly used for pets such as dogs, cats, horses, and pet pigs [13,22,23,24].

Laboratory pigs are increasingly used in experimental studies as they present important anatomical and physiological homologies with humans [25]. Pigs are cognitively and emotionally complex [26], but nevertheless many laboratory animal facilities only use habituation techniques to “train” the pigs in husbandry handling and experimental procedures [27]. Luring may also be preferred as it seems to be an intuitive and simple positive reinforcement technique. The use of CT is not widely used, even though pigs are easily clicker trained to cooperate for experimental procedures such as blood sampling via a venous access port, face mask inhalation, injections, and multiple X-rays [26,28,29].

Studies on the effect of various training methods on pig welfare are missing. It remains to be shown whether clicker training or luring reduces stress and fear in laboratory pigs during medical procedures or fear-evoking behavioural tests, and whether there is a difference between the two. The present study addresses the latter question.

Training an animal prior to an experimental procedure has the potential to increase animal welfare for several reasons. First, providing an animal with predictability or even controllability in an experimental set-up is a well-established way to reduce the negative physiological and psychological effects of experimental stressors [30]. One way of providing predictability and in some cases control over an experimental procedure is by training the animals using positive reinforcement. It is generally accepted that if a stressor is predictable in any form, it will have an anxiety-preventing effect [31,32]. Moreover, a lack of control over painful or otherwise stressful events has been shown to lead to depression and learned helplessness, defined as a debilitating cognitive state. It results in individuals who often possess the requisite skills and abilities to perform a task, but exhibit suboptimal performance [32,33]. It can therefore be suggested that controllability will induce the opposite condition, namely empowerment, defined as a cognitive state that results in increased intrinsic task motivation [33]. Using positive reinforcement to train cooperation during aversive (or potentially aversive) events will make these events more predictable for the pigs. Moreover, if all potentially aversive events have been included in the training, the pigs will also know that when the trainer has not cued, for example, “standing still for injection”, then no injections will be given. Hence, the non-appearance of this cue will function as a safety signal [30], leading to increased welfare. The use of PR will also give the pigs some control over the events. When the trainer gives a cue, the pig is presented with a choice. She can choose to show the behaviour the trainer asks for, or she can abstain from doing so—and in that respect she is in control of the events. It is obvious, then, that the trainer must accept if the animal decides not to participate. If the trainer has planned the session carefully (e.g., made sure that all environmental conditions are optimal and the pig is motivated to perform the target behaviour) then the pig will choose to work with the trainer and show the response wanted by the trainer [14,17,18]. Hence, it can be argued that using PR may reduce the negative effects of environmental stressors related to experimental procedures, medical procedures, and behavioural tests, thereby increasing animal welfare and staff safety.

Second, animals will continuously evaluate their environment as more or less comfortable [34], and positive emotions have been shown to be enhanced in specific situations. In pigs, the signalling of a reward (a preferred event or stimulus) will enhance positive emotions resulting in increased play behaviour and reduced aggression toward conspecifics [35]. Moreover, pigs who learned that a particular sound signalled that they could obtain food by pressing a button would show physiological changes indicating a positive mental state as a response to the sound signal [36]. It seems fair to suggest that using a clicker, for example, to signal the presentation of an appetitive reinforcer such as pieces of apple or raisins will also induce a state of positive anticipation in CT pigs [16]. Whether there is a difference between CT and luring at the level of positive anticipation during training sessions has not been scientifically addressed.

Third, during CT sessions, an improved human-animal relationship will arise, strengthening into a human-animal bond [37]. A human-animal bond is characterized as a relationship between a human and an individual animal; a relation which is reciprocal, persistent, and usually benefits both parties [38]. It can be suggested that this bond forms due to the building of trust between trainer and animal. The concept of “trust” is anthropomorphic in essence; however, it is intuitively understood by most humans and useful when evaluating the behavioural effects of animal training. The concept of “trust” is widely used in both pet animal training and zoos and aquariums [39]. In the laboratory animal society, it has been described by Poole (1997), who stated that “if the experimental animal has been trained to cooperate and has confidence and trust in the handler it will be much less stressed and the experiment will be much improved by the removal of this unwanted variable” [3]. Trust is a concept that can be defined as an expectation about future cooperation in contexts in which there is some incentive for partners to cheat [40]. It may also be defined as the reliance by an agent that actions prejudicial to their well-being are not undertaken by influential others [41]. In other words, trust is when you can rely on someone who is—in some way or aspect—stronger than you not to harm or cheat you, but to cooperate. Based on this interpretation of the concept of trust, it could be suggested that PR is actually building trust between the animal and the skilled trainer, as the pig will learn that the trainer consistently offers a possibility for the pig to obtain a preferred stimulus such as palatable food items. The skilled trainer will always set up conditions of the training session—including the criteria—in a way that maximizes the chance of the pig succeeding and hence obtaining the reinforcer. Using CT, the pig will also come to trust that the primary reinforcer always follows the click. When the skilled trainer sets up the conditions of the training correctly, then the training in itself will provide a safety signal, as the pig can predict any unpleasantness and choose not to participate.

In the following, we present a short, descriptive study on the behaviour of clicker trained or lured Göttingen mini pigs in a behavioural test, the novel task participation test (NTPT). The NTPT was inspired by a study on dolphins demonstrating that the willingness to participate in a training session is correlated with high animal welfare scores [42]. In short, the NTPT consists of a short runway with a novel walking surface not previously encountered by the pigs. Hence, this novel surface may be slightly fear-inducing when the pigs first step onto the runway. To assess the pigs’ willingness to participate in the task and the level of fearfulness, time taken to complete the task, number of trials needed to complete the task, number of primary reinforcers used, percent of task time spent tail wagging, and number of body stretches was measured (Table 1). Tail wagging behaviour has been shown to indicate a positive emotional state [43,44], and the duration of tail wagging has been found to be linked with play behaviour [45], a behaviour that has been suggested as an indicator of positive emotions and good welfare [46]. Body stretching is a risk-assessment behaviour shown as a stretched attend posture, when investigating potentially aversive areas. The pig stretches forward with an elongated body and both hind legs in a fixed position. This behaviour is well described in rodent literature [47,48], but this is not the case for pigs. However, as most pig caretakers will recognize the posture, we included it in this study.

It was expected that CT animals would be more willing to participate in the NTPT, confidently follow the trainer, and be less fearful of the novel walking surface. Hence, the CT pigs would complete the task faster, need fewer primary reinforcers, and show more tail wagging and fewer to no body stretches indicating more positive mental state and a lower level of anxiety.

## 2. Materials and Methods

### 2.1. Animals and Housing

Ten pigs participated in the novel task participation test (NTPT). They were intact, healthy, and well-socialized female Göttingen mini-pigs aged 11–13 months and weighing 22–26 kg. Upon arrival from the breeder, the pigs completed an acclimatization period of at least three weeks, in which they were housed in groups of four or six animals. After the acclimatization period, the pigs were single housed and randomly allocated to either the clicker training group (CT) or the luring group (LU). Two of the clicker-trained pigs had to be excluded from the study because the video recordings of the test were accidently lost, resulting in a total of three CT pigs and five LU pigs being included in the study. During single housing, snout-to-snout contact was possible through holes in the walls of the pens. The pigs were fed mini-pig diet (Special Diets Services (SDS), Essex, UK) twice a day for the first two weeks after arrival. Later, the pigs were fed once a day. All pigs had continuous access to fresh water, straw, wood shavings, and hay. All pens were cleaned every morning and provided with fresh hay, straw, and wood shavings. All stables were ventilated and kept at a temperature of 20 °C with a humidity of approximately 40%. In addition to daylight, artificial lights were turned on from 06:00 h to 18:00 h.

Enrichment items such as dumb bells, chains, and balls were provided and rotated between the pens. The pigs were also fed a cultured dairy product A38 (Arla foods, Viby J, 8260-DK), pieces of apples, and whole raisins on a daily basis in their pens and/or on the hallway floors when they were out for their daily exercise.

Socialization of the pigs was part of the husbandry procedure. During the daily exercise in the hallways of the facility and during pen cleaning, the animal caretaker would gently interact with the pigs and occasionally provide additional enrichment such as an empty food paper bag or an empty cardboard box. It should be noted that luring pigs to complete a task (for example, using pieces of apple to have them walk onto a scale for weighing and then back into their pen) could also be part of a standard husbandry procedure. As a result, all the pigs in this study were familiar with the principle of luring (“follow the food to get the food”).

Additionally, socialization was done according to a planned schedule from arrival and throughout the study. Scheduled socialization periods lasted approximately 10 min per session for group housed pigs and approximately 2 min for single housed. During a scheduled socialization period, one or two staff members would be present in the housing pen, evolving from just the presence of people in the pen, to the staff members sitting on the floor petting and scratching the pigs all over their bodies. The socialization periods were adjusted over time to fit the individual pigs so that pigs who were more cautious would eventually spend more time with staff members than the bolder ones. All ten pigs were comfortable with human contact at the time of their participation in the novel task participation test.

### 2.2. Clicker Training and Luring

The CT-group was trained to follow a target stick using pieces of apple as the primary reinforcer and a clicker as the conditioned reinforcer. Conditioning of the clicker was done by clicking and feeding a piece of apple ten times prior to target presentation. Conditioning was confirmed by observing the behaviour of the pig when the clicker was sounded, e.g., behind the pig. If the pig reacted to the sound with anticipatory behaviour, e.g., by approaching the trainer to receive the primary reinforcer, the clicker was considered conditioned. Throughout the entire study, the conditioned reinforcer was always followed by the primary reinforcer. The behaviour (“follow target stick”) is a simple behaviour that is very easy for a well-socialized pig to do as they are curious and confident and will explore the target readily, if it is presented strategically by the trainer. The trainer captured the “touch target” behaviour and when the pig consequently touched the target when it was presented, the target was moved away from the pig, prompting the pig to follow it to touch. The “follow target” behaviour was then reinforced (no longer the “touch target” behaviour) and the time required to follow to obtain the reinforcer was stepwise increased. Pigs who were more hesitant were trained to touch the target using shaping (i.e., successive approximations towards the final goal behaviour) during which, for example, the following criterias were used: first “looking at target”, then “orienting towards target”, “moving towards target”, and last “touching target”. The training method (shaping or capturing) was decided by the trainer based on the trainer’s knowledge of the individual pig and the pigs’ behaviour in the training situations. The same trainer (who also did the novel task participation test) trained all CT pigs for four weeks. Depending on the husbandry plans for the day, pigs received up to three training sessions per day. During this period, the pigs were trained to follow the target, go onto a scale or a raised platform, stand still for injections and oral dosing, and stand still for X-rays. One training session lasted a maximum of 6 min. The pigs in the LU group were socialized and fed pieces of apple for a similar amount of time. The LU pigs were trained using luring to follow the trainer and to, e.g., go onto a scale; however, no purposive training for injection or other potentially aversive procedures was done. Consecutive days without training did not exceed five days. Most training sessions and all novel task participation tests (NTPTs) were performed in the mornings before the pigs were fed. All CT pigs were thus trained for several behaviours prior to the NTPT.

### 2.3. Novel Task Participation Test (NTPT)

A novel task was designed by constructing a novel walking surface using three plastic grass turf mats (Clean Carpet Finnturf^®^) with rigid brushes measuring 60 × 90 cm and covered with transparent plastic bags, providing a surface none of the pigs had ever encountered before. This “novel floor” was placed in a test room not previously known to the pigs. It was not cleaned between trials. The order of testing was randomized between all ten pigs. The animal trainer guided the CT pigs from the home pen into the test room, over all three mats and back to the home pen using the familiar target stick. The LU group was guided through the same sequence using luring (encouraging calls and pieces of apple) by the same trainer. At any given time, it was possible for the pig to choose not to participate and simply return to her pen. All pigs chose to participate.

### 2.4. The Test Variables

During the NTPT, the variables presented in Table 1 were assessed. The NTPT started when the pig placed the first cloven hoof on the first mat and ended when the last back hoof was lifted from the third mat. Test variables (Table 1) during the NTPT were determined using video recordings. The recordings were performed by a person known by the pigs. This person was located in the room from the start of the test to reduce disturbance. The videos were analysed by two different people. As the CT pigs were guided through the task using a target stick, whereas the LU group were lured, is was not possible to blind the video-observers. No coding or software was used to analyse the videos. The videos were run in slow-motion and time-points for the start and end of the predefined behaviours were noted. Interrater reliability was not calculated, as the two observers were in agreement.

### 2.5. Statistics

Due to the small sample size in the CT group, only descriptive data on individual animals are presented. Descriptive statistics (median and interquartile range s IQR) were calculated using GraphPad Prism version 8 (GraphPad Software, San Diego, CA, USA).

### 2.6. Ethical Considerations

The pigs were not purchased for the purpose of the novel task participation test; they were all part of a larger study scheduled at the facility. As no negative effects were expected from the clicker training and the luring, this clicker training study was done prior to the main study for which the pigs were originally obtained. No animal experimentation license was needed for the study as the CT/luring and NTPT were not considered a procedure as defined in the EU directive 2010/63/EU; article 3.

## 3. Results

Five LU pigs (ID no. 2, 6, 8, 9, and 10) and 3 CT pigs (ID no. 1, 3, and 4) were included in the study. Due to technical issues, two CT pigs (ID no. 5 and 7) had to be excluded from the study, and as a result no statistical analysis of the data was done due to the low sample size in the CT group and an overall large variation. Five variables were assessed during the NTPT, four of which are presented as descriptive data (Figure 1a–d).

### 3.1. Time and Number of Attempts to Complete the NTPT

All pigs completed the task relatively quickly, ranging from 3 s (CT pig no. 4) to 22 s (LU pig no. 2) (Figure 1a). One CT pig and one LU pig (no. 1 and 2) used two attempts to complete the task, as they stepped onto the mat, stepped off the mat, and then back onto the mat again (data not shown).

### 3.2. Primary Reinforcers

The number of apple pieces used by the trainer to guide the pigs through the NTPT ranged from 1 (CT pig no. 4) to 8 (LU pig no. 2) (Figure 1b).

### 3.3. Tail Wagging and Body Stretches

Tail wagging is presented as percent of total time used to complete the task (Figure 1c), as the time to complete the task varied among the pigs. Interestingly, all CT pigs showed some tail wagging during the task (ranging from 33% to 38%), whereas the lured pigs—with one exception—showed none. One LU pig (no. 10) showed tail wagging 56% of the time during the task, which exceeded the level of tail wagging in the CT pigs. Only one pig (LU pig no. 2) displayed body stretching when performing the task (Figure 1d).

## 4. Discussion

All pigs completed the task. However, LU pig no. 2 and LU pig no. 10 showed an overall behavioural profile different from that of the remaining pigs. Pig no. 2 was taking more than twice the time to complete the task compared to the other LU pigs. During the task, she needed more reinforcers than the other pigs (which is not surprising as she was at it for a longer time), and during the task she showed two body stretches (a behaviour not shown by any of the other pigs) and no tail wagging. Overall, pig no. 2 presented as more sensitized and anxious in the test situation than the other pigs. LU pig no. 10, on the other hand, displayed a lot of tail wagging, especially compared to the other four lured pigs (who all showed none). As tail wagging has been shown to indicate a positive state of mind [43,44,45], it could be argued that the CT pigs and LU pig no. 10 were in a more positive state of mind compared to the other four LU pigs. Moreover, tail wagging has been reported to be related to eating in a positive situation [49], so the feeding of apples may have increased the positive valence (as perceived by the pigs) of the NTPT and for some reason especially so in pig no. 10. As all pigs completed the task, future studies should consider training pigs for more aversive procedures such as injections or blood sampling, which are more difficult to train using luring compared to CT. A behaviour that is more difficult for the animal to do (such as standing still for injections) could to a higher extent reveal the difference between luring and clicker training. Obviously, both methods (LU and CT) must then be used at their best.

All the pigs in this study were socialized to a high degree, and all were confident around humans, readily accepted treats, and followed the handler willingly. During the design-phase of the study, it was discussed whether a non-socialized group of clicker-trained pigs should be included. However, as the facility in which the study was done never performs studies on non-socialized pigs for animal welfare reasons, it was decided to omit a non-socialized group of pigs and focus on the effect of clicker training versus luring in socialized pigs. However, in future studies, including a non-socialized, clicker trained group could be interesting to establish the full potential of the CT method. Luring non-socialized pigs would be a complicated task, but of course, such a group could be included as well.

The LU pigs were not exposed to a similar amount of novelty during training as were the CT pigs. LU pigs were—due to the nature of the training method—not trained to go onto raised platforms, accept injections, and accept oral dosing, as this could risk inducing fear of the trainer or of the training environment. This difference in “novelty-training” may have reduced the risk of sensitization due to novelty during testing in CT pigs compared to LU pigs, and hence made the CT pigs comparatively more confident during the test. We did expect the CT pigs to be more confident simply due to the training method; however, there is a risk that part of the difference is due to the above-mentioned difference in training prior to testing. A focus-point in future studies should therefore be the use of more comparable training sessions in CT and LU pigs prior to testing.

In conclusion, training pigs using PR in the form of CT or luring resulted in pigs that willingly performed a novel and potentially fear-evoking task, namely walking over a novel surface. None of the pigs showed behavioural signs of fear such as running away. One LU pig showed a combination of body stretching and no tail wagging indicating anxiety, possibly because of a high level of sensitization due to the novelty of the test situation, and the absence of the positive mental state relating to tail wagging. All three CT pigs and one out of five LU pigs showed tail wagging indicating a positive state of mind. Overall, training pigs prior to an experimental, potentially fear evoking behavioural test using either CT or LU will result in pigs that quickly, voluntarily, and without the need to apply any force, complete the task. Future studies should aim to demonstrate that using positive reinforcement training methods for training cooperation during experimental procedures will reduce stress and anxiety in animals. This will, in turn, result in enhanced animal welfare, as well as increased quality of data.

## Figures and Tables

**Figure 1 animals-11-01610-f001:**
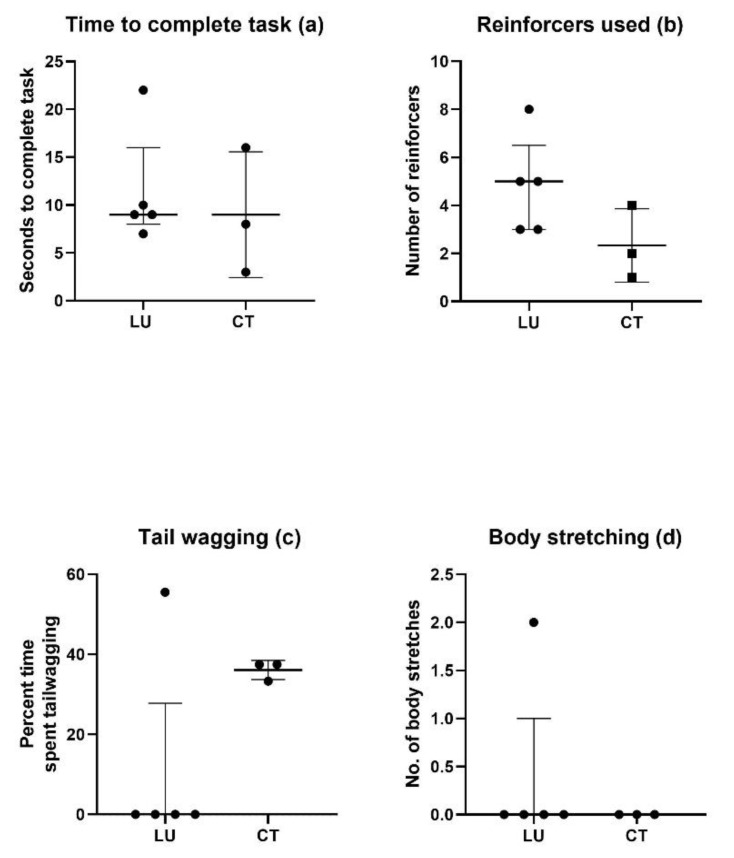
Variables assessed during the task (i.e., walking across the three plastic-coated plastic turf mats). Each individual pig is presented. Median and IQR (interquartile range) is shown on all four figures. (**a**) The time (seconds) it took each pig to complete the task. (**b**) The number of reinforcers the trainer used during the task. (**c**) Tail wagging during the task shown as percent of time spent tail wagging. (**d**). Body stretching. The number of body stretches shown by each pig during the task is shown. LU: luring, CT: Clicker training.

**Table 1 animals-11-01610-t001:** Variables assessed during the Novel Task Participation Test (NTPT).

Variable and Unit of Measurement	Description
Time to complete task (s)	The time from task start (first front cloven hoof is placed on the first mat) to when the pig stepped off the mat (last back cloven hoof was lifted from the last mat).
Number of attempts to complete task	Number of times the pig stepped onto the mat before finishing the task.
Number of primary reinforcers (apple bites) used to complete task	Number of pieces of apple fed by the handler during the task.
Tail wagging (percent of “time to complete task” showing tail wagging)	Percent of time spent tail wagging (tail swinging in any direction, but mostly from side to side) during the full duration of the task.
Body stretching (number of body stretches during the task)	Body stretching is an exploratory posture, indicating anxiety/uncertainty, in which the pig stretches forward with both hind legs in a fixed position while investigating a potentially aversive area.

## Data Availability

The behavioural data presented in this study are available on request from the corresponding author. The data are not publicly available due to practical reasons. Videos are available from the corresponding author with a joint permission from Novo Nordisk Vice president Jan Lund Ottesen and the corresponding author.

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
