# Peer review of "A Case Study on the Behavioural Effect of Positive Reinforcement Training in a Novel Task Participation Test in Göttingen Mini Pigs"

_animals, 2021, doi:10.3390/ani11061610_

Round 1

Reviewer 1 Report

In this extraordinarily simple study, the experimenters lured pigs across a novel walking surface using either pieces of apple or a clicker that had previously been paired with pieces of apple. They found, unsurprisingly, that pigs had a generally positive response to this and were willing to walk across a runway with novel objects and surfaces on it. I found this paper completely unsurprising, as I am sure the authors did too, since I already believed that animals will follow food and food-associated cues. The data is too limited for statistical analysis, but are those really necessary when the results are so obvious? I could criticise other things in this paper too, like the unbalanced and tiny sample sizes or the lack of a real experimental procedure that the pigs were being introduced to, but I just don't think those are necessary to prove that intelligent animals like pigs will follow apple pieces or apple-associated cues.

I think the paper was well written. My main suggestion is to acknowledge that the discussion of trust on lines 130-152 is potentially an anthropomorphism - though the authors do attempt to define trust as based on expectation. My only other nitpicks are to suggest that clicker training must have a better reference than clickertraining.com [18] - perhaps something in an academic journal, and also to ask the authors to define their error bars in the figure legend (SD I presume, based on the methods).

Author Response

Reviewer 1

In this extraordinarily simple study, the experimenters lured pigs across a novel walking surface using either pieces of apple or a clicker that had previously been paired with pieces of apple. They found, unsurprisingly, that pigs had a generally positive response to this and were willing to walk across a runway with novel objects and surfaces on it. I found this paper completely unsurprising, as I am sure the authors did too, since I already believed that animals will follow food and food-associated cues. The data is too limited for statistical analysis, but are those really necessary when the results are so obvious? I could criticise other things in this paper too, like the unbalanced and tiny sample sizes or the lack of a real experimental procedure that the pigs were being introduced to, but I just don't think those are necessary to prove that intelligent animals like pigs will follow apple pieces or apple-associated cues.

 I fully agree; the results are not surprising at all. However, working in lab animal science I repeatedly see that researchers do not allocate resources to train the pigs, even though the benefits on animal welfare are obviously high. So we decided to show how easily you can make a pig walk a novel and potentially scary surface using positive reinforcement techniques in the hope of making the usefulness of these training techniques obvious to those not using these tools yet. I have emphasized this point by changing the followed section from: " Pigs are cognitively and emotionally complex [22] and in many laboratory animal facilities, pigs are trained using habituation or luring [23]. However, pigs are easy to clicker train to cooperate for example during blood sampling via a venous access port, face mask inhalation, injections and multiple x-rays [22,24,25]" to " Pigs are cognitively and emotionally complex [22], but never-the-less many laboratory animal facilities only use habituation techniques to "train" the pigs to husbandry handling and experimental procedures [23]. Luring may also be used as it seems to be an intuitive and simple positive reinforcement technique. The use of CT is not widely used even though pigs are easily clicker trained to cooperate for experimental procedures such as blood sampling via a venous access port, face mask inhalation, injections and multiple x-rays [22,24,25].

It is of course most inconvenient that we lost data due to technical issues; however, we decided to submit our study anyway as we find it necessary to draw attention to the trainability of laboratory pigs.

I think the paper was well written. My main suggestion is to acknowledge that the discussion of trust on lines 130-152 is potentially an anthropomorphism - though the authors do attempt to define trust as based on expectation.

I agree; "trust" is difficult to define and show in animals. I have therefore made the following addition: "It can be suggested that this bond will form due to the building of trust between trainer and animal. The concept of "trust" is anthropomorphic in essence; however, it is intuitively understood by most humans and useful when evaluating the behavioural effects of animal training. The concept of “trust” is widely used in both pet animal training and Zoos and aquariums"

My only other nitpicks are to suggest that clicker training must have a better reference than clickertraining.com [18] - perhaps something in an academic journal,

It was not possible (for me, at least :-)... ) to find discussions on the difference between targeting and luring in an academic journal as these techniques are practical animal trainer techniques. I have, however, added an additional reference on clicker training after the paragraph: "Clicker training (CT) builds on a learned (conditioned) association between a previously neutral stimulus such as a whistle or a clicker (small device that, when pressed, gives a distinct “click-click” sound”) and a primary (or unconditioned) reinforcer, typically food. When the animal shows the correct response, the conditioned reinforcer is presented followed by the primary reinforcer (15,19) (ref 19: Dorey and Cox (2018), Function matters: a review of terminological differences in applied and basic clicker training research. PeerJ 6:e5621; DOI 10.7717/peerj.5621

and also to ask the authors to define their error bars in the figure legend (SD I presume, based on the methods).

As the data set is very small and one of the other reviewers therefore requested that we showed the data with median and ICR instead of mean and SD, we chose to provide this. I hope it is acceptable.

Reviewer 2 Report

This is an interesting paper. It makes original contributions to knowledge about incorporating clicker training (CT) in research pigs in a lab setting. However, a big concern will be the small sample size, indicating a lack of solid conclusion. Also, luring (LU) has already been used, and there is a lack of evidence in this study showing that CT is more friendly to animal welfare or more effective than LU. Authors may want to explain more about the importance of this study.

Introduction

  • Line 64: ‘such as for example luring’. You may say either ‘such as’ or ‘for example’. Please correct it.
  • Line 83: ‘Never the less’. Please delete the spaces.
  • Line 127: ‘in this aspect…’. It is a bit confusing. In what aspect are you referring? Please rephrase.
  • Line 166-170: ‘Body stretching is a risk-assessment behaviour…in this study’. Would you please provide more references and solid arguments to justify that the same behaviour found in rodents can be generalised to pigs? Otherwise, it may not be suitable to conclude that this behaviour is a stress-related signal in pigs.
  • Table 1 (Number of primary reinforcers (ap- ple bites) used to complete task): Is the number of primary reinforcers used same as that of secondary (conditioned) reinforcer (clicker) used?
  • Table 1 (Tail wagging): ‘percent of time to complete task showing tail wag- ging’. This sentence is a bit confusing. Please rephrase it.

Materials and Methods:

  • Line 225-226: ‘Conditioning of the clicker was done by clicking and feeding a piece of apple ten times prior to target presentation.’ How did you know that the pigs had linked the conditioned reinforcer with the primary reinforcer?
  • Line 246: ‘All CT pigs were thus trained for several behaviours prior to the NTPT’. Were CT pigs trained to display behaviours other than those described in Line 224-243? If yes, what behaviours? And how these trained behaviours might affect their presentations during the NTPT?
  • Line 267: ‘The videos were analysed by two different persons’ What is the inter-rater reliability between two video coders? Also, what was the coding method? Did they code the videos in their entirety? With what software?
  • Line 277-278: ‘The pigs were not purchased for the purpose of the Novel Task Participation Test; they were all part of a larger study scheduled at the facility.’ What was the previous research history of these pigs? Did they have the same research history? How might the research history affect your study?

Results:

  • Figure 1: Due to the small sample size, the data was not likely to be normally distributed. Therefore, please also provide the median and interquartile range.

Discussion:

  • Line 318: Please define the ‘deviant behaviour’.

Conclusion:

In this study, is there any sign showing that CT may be better the LU (i.e. more effective, more welfare-friendly, easier to implement)? If LU has already been used and is similarly friendly to animal welfare and is not more time-consuming than CT, why CT should be encouraged? What is the impact of this study and how this study can help scientists and research animals?

Author Response

Reviewer 2

This is an interesting paper. It makes original contributions to knowledge about incorporating clicker training (CT) in research pigs in a lab setting. However, a big concern will be the small sample size, indicating a lack of solid conclusion. Also, luring (LU) has already been used, and there is a lack of evidence in this study showing that CT is more friendly to animal welfare or more effective than LU. Authors may want to explain more about the importance of this study.

We thank you for your comments, suggestions and the insightful questions to the actual application of the two training methods.  We have addressed them all point-by-point below.

Introduction

Line 64: ‘such as for example luring’. You may say either ‘such as’ or ‘for example’. Please correct it. "for example" has been deleted.

Line 83: ‘Never the less’. Please delete the spaces. "Never-the-less" has been deleted in the review-process.

Line 127: ‘in this aspect…’. It is a bit confusing. In what aspect are you referring? Please rephrase.

The sentence has been re-phrased to: "Whether there is a difference between CT and luring in the level of positive anticipation during training sessions has not been scientifically addressed."

Line 166-170: ‘Body stretching is a risk-assessment behaviour…in this study’. Would you please provide more references and solid arguments to justify that the same behaviour found in rodents can be generalised to pigs? Otherwise, it may not be suitable to conclude that this behaviour is a stress-related signal in pigs.

I fully understand your point and I, too, would have preferred to have published papers describing the behaviour in pigs. However, the body stretching behaviour is to my knowledge not described in any published pig ethograms. But as it is a behaviour well-known to pig care takers, we decided to include it. The behaviour seems to be identical to that of mice; both from a locomotor perspective and from the similarities of the situations in which the behaviour is displayed. We have therefore described the behaviour as an exploratory behaviour seen in novel and potentially fear-evoking environments.

Table 1 (Number of primary reinforcers (ap- ple bites) used to complete task): Is the number of primary reinforcers used same as that of secondary (conditioned) reinforcer (clicker) used?

Yes. We always train using the "click-treat" principle. This is a very good point as different trainers may have different approaches. To clarify, we have therefore added the following to Material and methods, 2.2: "Throughout the entire study the conditioned reinforcer was always followed by the primary reinforcer."

Table 1 (Tail wagging): ‘percent of time to complete task showing tail wag- ging’. This sentence is a bit confusing. Please rephrase it.

Good point. For clarification, I have changed it to: " Tail wagging (percent of "time to complete task" showing tail wagging)" and added: "Percent of time spent tail wagging (tail swinging in any direction, but mostly from side to side) during the full duration of the task.

Materials and Methods:

Line 225-226: ‘Conditioning of the clicker was done by clicking and feeding a piece of apple ten times prior to target presentation.’ How did you know that the pigs had linked the conditioned reinforcer with the primary reinforcer?

Conditioning was tested as the trainer observed the behaviour of the pig, when the clicker was sounded e.g. behind the pig. If the pig reacted to the sound with anticipatory behaviour, e.g. by approaching the trainer to receive the primary reinforcer, the clicker was considered conditioned. The clicker was never used without being followed be the primary reinforcer.

Line 246: ‘All CT pigs were thus trained for several behaviours prior to the NTPT’. Were CT pigs trained to display behaviours other than those described in Line 224-243? If yes, what behaviours? And how these trained behaviours might affect their presentations during the NTPT?

All pigs were trained to follow target out of the pen and onto a scale. I have added this to the description. Also, Reviewer 3 had comments to this point, so we have added the following (copy-pasted from answers to reviewer 3): "I have clarified the following to section 2.2.: " The LU pigs were trained using luring to follow the trainer and e.g. go onto a scale; however, no purposive training for injection or other potentially aversive procedures was done.".

Moreover, I have expended the discussion including the following paragraph:" The LU pigs were not exposed to a similar amount of novelty during training, as were the CT pigs. LU pigs were – due to the nature of the training method – not trained to go onto raised platforms, accept injections and accept oral dosing as this could risk inducing fear of the trainer or of the training environment. This difference in "novelty-training" may have reduced the risk of sensitization due to novelty during testing in CT pigs compared to LU pigs and hence made the CT comparatively more confident during the test. We did expect the CT pigs to be more confident simply due to the training method; however, there is a risk that part of the difference is due to the above-mentioned difference in training prior to testing. A focus-point in future studies should therefore be the use of more comparable training sessions in CT and LU pigs prior to testing".

Line 267: ‘The videos were analysed by two different persons’ What is the inter-rater reliability between two video coders? Also, what was the coding method? Did they code the videos in their entirety? With what software?

No coding or software was used to analyse the videos. The videos were run in slow-motion and time-point for start and end of the predefined behaviours were noted. We did not calculate the inter-rater reliability as the two observers were in agreement.

Line 277-278: ‘The pigs were not purchased for the purpose of the Novel Task Participation Test; they were all part of a larger study scheduled at the facility.’ What was the previous research history of these pigs? Did they have the same research history? How might the research history affect your study?

None of the pigs had been subjected to any procedures prior to training. The training and the Novel Task Participation Test was done before the pigs were used for further research projects. Basically, we were simply allowed to socialize, train and test the pigs in a slightly extended acclimatisation period.

Results:

Figure 1: Due to the small sample size, the data was not likely to be normally distributed. Therefore, please also provide the median and interquartile range.

 We fully agree that the data is not normally distributed and the sample size very small. We have therefore chosen to present the study as a case study showing all individual data points in the figure. We – the authors – did discuss intensively whether to present median and interquartile range or mean and SD – or simply just present the data points.

We have made the figure with median and interQR as requested – both figures are attached for comparison.

Discussion:

Line 318: Please define the ‘deviant behaviour’.

The sentence has been clarified: " an overall behavioural profile different from that of the remaining pigs".

Conclusion:

In this study, is there any sign showing that CT may be better the LU (i.e. more effective, more welfare-friendly, easier to implement)? If LU has already been used and is similarly friendly to animal welfare and is not more time-consuming than CT, why CT should be encouraged? What is the impact of this study and how this study can help scientists and research animals?

 Luring is often used to move pigs from one place to another or the like; and the LU pigs in this study was also trained to do so. However, behaviours that are more difficult for the pigs (such as standing still for injections) are better trained using CT. Unfortunately, the chosen test in this study does not reflect this difference.

The following paragraph has therefore been added to the discussion: " As all pigs completed the task, future studies should consider training pigs for more aversive procedures such as injections or blood sampling, which are more difficult to train using luring compared to CT. A behaviour that is more difficult for the animal to do (such as standing still for injections) could to a higher extent reveal the difference between luring and clicker training. Obviously, both methods (LU and CT) must then be used at its best."

Moreover, to emphasize the point on how animal training can help both scientists and animals, I have added the following to the final discussion:  "Future studies should aim to demonstrate that using positive reinforcement training methods for training cooperation during experimental procedures will reduce stress and anxiety in the animals. This will in turn result in enhanced animal welfare as well as increased quality of data." 

Reviewer 3 Report

I have some suggestions please see attached document

Author Response

Reviewer 3

This is a really nice paper, and an important topic. I have a few suggestions and apologise for delay. My suggested changes are in bold.

We thank you for the insightful and valuable comments .We have corrected the text according to all the suggestions and we have added the suggested references in the relevant paragraphs.  Also, we have carefully addressed the requests on expanding and clarifying some of the sections. Below, we have addressed all comments point-by-point; we have divided the comments into "simple suggestions"; "suggestions on references" and "Clarifications/expansions".

Simple suggestions

Line 43 Negatively effects animal physiology and psychology and thereby…. Corrected accordingly.

Line 44 …. research data [1-3]. Major sources of stress are….. Corrected accordingly.

Line 45 replace fixated with restrained Corrected accordingly.

Line 49 behavioural test should be behavioural tests Corrected accordingly.

Line 54 comma between stressed and when needs deleting Corrected accordingly.

Line 84 …methods on pig welfare are missing. It remains to be shown…. Corrected accordingly.

Line 92 physiological and psychological effects Corrected accordingly.

Line 97 cognitive state. This results in individuals who often possess Corrected accordingly.

Line 115-116 increase animal welfare, and staff time and safety.  Corrected accordingly.

Line 130 human-animal relationship Corrected accordingly.

Line 146 comma between learn and that needs deleting Corrected accordingly.

Line 180 and fewer to no body stretches Corrected accordingly.

Line 194 feed should be fed Corrected accordingly.

Line 213 socialization periods lasting approximately Corrected to "lasted approximately"

Line 168 when investigating potentially aversive areas. Corrected accordingly.

Table 1 Body stretching I suggest needs clarity of description. It indicates a degree of anxiety.. not merely exploration and it is not investigating an ‘aversive’ area, but an area that is ‘potentially aversive’ I suggest therefore Body stretching is an exploratory posture, indicating anxiety / uncertainty, in which…… investigating a potentially aversive area… Corrected accordingly.

Line 233 Pigs who were more hesitant, were… Corrected accordingly.

Line 319 reinforcers than the … Corrected accordingly.

Suggestions on References

Line 44 [1-3] can I also suggest you add Olsson, I. A. S., Nevison, C. M., Patterson-Kane, E. G., Sherwin, C. M., Van de Weerd, H. A., & Würbel, H. (2003). Understanding behaviour: the relevance of ethological approaches in laboratory animal science. Applied Animal Behaviour Science, 81(3), 245-264.

This reference has been added.

Line 76 cats, horses and pet pigs [12, 20] and I suggest Kurland book is replaced by Hart, B. (2008). The art and science of clicker training for horses: a positive approach to training equines and understanding them. Souvenir Press. You may also wish to add Chronister, K. and Matlock, S 2016 Mini Pig Training Handbook: Tricks, Life Skills, and Communication with Your Mini Pig CreateSpace Independent Publishing Platform

The pet pigs have been added to the text and the references included. I thank you for directing my attention to these two books. They are not available in literature searches at the Copenhagen University library (or any other Danish library), so I was not aware of them – until now J

Clarifications/expansions

Line 72-73 needs a little more for clarity.. I suggest… This training approach facilitates the refinement of the desired behaviour offered by the animal for example in its precision or timing. This method has been applied over the years……..

I see your point. I have tried to clarify by rewording to: "This training approach will increase the trainers' timing as the behaviour desired by the trainer can be marked with high precision using the clicker (ref : Sørensen, D.B.; Pedersen, A.; Bailey, R.E.B. Animal Training: The Practical Approach. In Animal-centric Care and Management - Enhancing Refinement in Biomedical Research, Sørensen, D.B., Cloutier, S., Gaskill, B.N., Eds. CRC Press. Taylor and Francis Group: Boca Raton, Florida, 2021; pp. 73-90.)".

Line 119 and throughout paper please replace ‘positive’ emotions with either appetitive or pleasant. It saves any confusion with the term positive as used in learning theory (i.e. positive reinforcement or positive punishment).

This is a very interesting point and one we – the authors – have discussed intensively prior to submission – and for the exact same reasons. The challenge is that both positive reinforcement as well as positive emotions (or positive affective states) are both well-established concepts (see for example papers by Reimert et al and Boissy et al in references) and I would therefore prefer to keep the original wording.

Line 344… I do not agree with the lack of nuance in this conclusion. No tail wagging may not have indicated fear but certainly indicated anxiety and sensitisation.. which means a fear association is more likely to be made and only may become apparent at a future time in the same / similar situation. Further, anxiety of itself may affect data.

I would like to see the authors to expand on this in this paragraph.

This is a very good point. I agree that the absence of tail wagging is not indicating fear, but merely a lack of the positive emotions relating to the behaviour. The point on using anxiety and sensitization instead of fear makes very good sense – these pigs are not afraid; they are sensitized due to the novelty and slightly anxious. I have made the following changes:

Abstract/summary: "Hence, training pigs to cooperate during experimental procedures resulted in a smooth completion of the task with no signs of fear or anxiety in 7 out of 8 animals"

Last word in introduction is changed: "CT pigs would complete the task faster, need fewer primary reinforcers and show more tail wagging and fewer to no body stretches indicating more positive mental state and a lower level of fear anxiety".

Discussion: " Overall, pig no. 2 presented more fearful than the other pigs" was changed to" Overall, pig no. 2 presented more sensitized and anxious in the test situation than the other pigs."

Conclusion: "Most pigs showed no behavioural signs of fear, but only CT pigs and one out of five LU pigs showed tail wagging behaviour indicating a positive state of mind. One LU pig showed the combination of no tail wagging and body stretching indicating a more fearful state." has been changed to "None of the pigs showed behavioural signs of fear such as running away. One LU pig showed a combination of body stretching and no tail wagging indicating anxiety, possible because of a high level of sensitization due to the novelty of the test situation, and the absence of the positive mental state relating to tail wagging. Only CT pigs and one out of five LU pigs showed tail wagging indicating a positive state of mind".

Further, there is a confounding variable in the methods that has not been addressed. The CT group had a lot more experience of novel situations through their pre test training, and of ‘trusting humans’ in a training situation. This of itself may have increased confidence. Why were control group not lure trained to a comparable number of different tasks? Or at least exposed to some degree of novelty? Can the authors please address this potentially large confounding variable and its potential effect on the data and suggest future refinements.

I fully agree that this constitutes a confounding factor – especially the lack of exposure to novelty. The LU pigs were trained to follow the trainer out of the pen and onto a scale and back to the pen; however, as we decided that luring the pigs onto a platform and luring the pigs to accept injections could induce fear of the trainer/training situation, we did not try to do it. From a practical training perspective, the correct technique for training to cooperate during aversive procedures is CT, and we decided to only subject the pigs to training that was relevant considering the training method (CT or LU). However, it is a very valid point that the training prior to testing should be comparable.

I have clarified the following to section 2.2.: " The LU pigs were trained using luring to follow the trainer and e.g. go onto a scale; however, no purposive training for injection or other potentially aversive procedures was done.".

Moreover, I have expended the discussion including the following paragraph:" The LU pigs were not exposed to a similar amount of novelty during training, as were the CT pigs. LU pigs were – due to the nature of the training method – not trained to go onto raised platforms, accept injections and accept oral dosing as this could risk inducing fear of the trainer or of the training environment. This difference in "novelty-training" may have reduced the risk of sensitization due to novelty during testing in CT pigs compared to LU pigs and hence made the CT comparatively more confident during the test. We did expect the CT pigs to be more confident simply due to the training method; however, there is a risk that part of the difference is due to the above-mentioned difference in training prior to testing. A focus-point in future studies should therefore be the use of more comparable training sessions in CT and LU pigs prior to testing".

Round 2

Reviewer 2 Report

  • Please add the detail (as stated in your response) regarding how you confirmed the pigs had linked the conditioned reinforcer with the primary reinforcer into the manuscript.
  • Please add the detail of your video coding method (as stated in your response) into the manuscript.

Author Response

Original comment from Reviewer 2:

Line 225-226: ‘Conditioning of the clicker was done by clicking and feeding a piece of apple ten times prior to target presentation.’ How did you know that the pigs had linked the conditioned reinforcer with the primary reinforcer?

We have included the following in the manuscript:

"Conditioning was confirmed by observing the behaviour of the pig, when the clicker was sounded e.g. behind the pig. If the pig reacted to the sound with anticipatory behaviour, e.g. by approaching the trainer to receive the primary reinforcer, the clicker was considered conditioned".

Original comment from Reviewer 2

Line 267: ‘The videos were analysed by two different persons’ What is the inter-rater reliability between two video coders? Also, what was the coding method? Did they code the videos in their entirety? With what software?

We have included the following in the manuscript:

"No coding or software was used to analyse the videos. The videos were run in slow-motion and time-points for start and end of the predefined behaviours were noted. Interrater reliability was not calculated as the two observers were in agreement."